# Machine Learning Applications for the Development of a Questionnaire to Identify Sasang Constitution Typology

**DOI:** 10.3390/ijerph191811820

**Published:** 2022-09-19

**Authors:** Soon Mi Kim, Jeongkun Ryu, Eunhye Olivia Park

**Affiliations:** Department of Food and Nutrition, College of BioNano Technology, Gachon University, Seongnam 13120, Korea

**Keywords:** Sasang constitutional medicine, machine learning, feature selection, hierarchical clustering analysis, Sasang constitution typology

## Abstract

Sasang constitutional medicine emphasizes personalized disease prevention and treatment and has been used in various fields. Nevertheless, more efforts are required to improve the validity and reliability of the Sasang analysis tools. Hence, this study aimed to (1) identify key constructs and measurement items of the Sasang constitution questionnaire that characterize different Sasang constitutions and (2) investigate the similarities and differences in pathophysiological and personality traits between Sasang constitutions. The results of the Sasang constitution questionnaire were analyzed using multiple machine learning-based approaches, including feature selection, hierarchical clustering analysis, and multiple correspondence analysis. The selected 47 key measurement items were clustered into six groups based on the similarity measures. The findings of this study are expected to be beneficial for future research on the development of more robust and reliable Sasang conservation questionnaires, allowing Sasang constitutional medicine to be more widely implemented in various sectors.

## 1. Introduction

In oriental medicine, the human body is viewed as a microcosm of the universe, and diseases are diagnosed and treated based on the yin-yang and five-element theories that health and longevity are achieved through a harmonious relationship between spiritual, mental, and physical states [1,2]. Despite sharing the same oriental philosophical roots as traditional Chinese medicine, Korean Sasang constitutional medicine has developed distinct methods for identifying and treating diseases [2,3]. Specifically, traditional Chinese medicine emphasizes exterior similarities of individuals, such as body type, temperament, symptoms, and pathology, in disease diagnosis and treatment [4]. On the other hand, in Sasang constitutional medicine, the major cause of the disease is an imbalance of emotions (e.g., sadness, anger, happiness, and joy) derived from human relationships and self-control; therefore, mind regulation is a key element for disease treatment [1,5]. 

The Sasang constitutional theory was established by physician Je-Ma Lee (1837–1900) at the end of the nineteenth century in his book *Donguisusebowon*, which classifies people into four constitutional categories: Tae-Yang (TY), Tae-Eum (TE), So-Yang (SY), and So-Eum (SE) [6]. According to Sasang constitutional theory, humans are born with a certain energy that influences their appearance, personality, and disease symptoms, and as a result, individuals’ minds and organs function differently [7]. Given that these individual differences may influence receptivity to a particular treatment and the efficacy of medicine, the Sasang constitutional theory suggests individualized disease preventive and treatment options [8]. Moreover, the Sasang constitutional theory underlines the importance of healthy lifestyles, such as positive attitudes and good relationships with others, in disease treatment and prevention [9]. 

The most reliable method for determining the Sasang constitution is to monitor improvements in symptoms with herbal medicine prescribed by Korean medicine doctors [10]. However, this method may not be easily adaptable to the Sasang constitution in various industrial domains. The Korea Institute of Oriental Medicine has developed the Sasang Constitution Analysis Tool (SCAT) to identify an individual’s Sasang constitutional types through four measures: facial type, voice, body type, and questionnaires [11]. The Questionnaire for Sasang Constitutional Classification II (QSCCII) is another popular form of the Sasang Constitution Questionnaire that consists of multiple constructs, such as body shape, health status, and eating behavior [12,13]. However, concerns have been raised about how accurately these methods determine Sasang constitutional types [14,15]. To implement holistic Sasang constitution determination methods, experts in the private sector have also developed techniques using pendulum and O-ring tests employing various means such as scent oils, color cards, gold and silver rings, and observation of individual faces and body shapes [16]. Previous studies [16,17,18,19] that employed both methods mentioned above (i.e., SCAT and field experts) revealed that the two methods produced different Sasang determination outcomes, demonstrating a low concordance rate between the two methods. 

Although self-reported surveys are widely used for the Sasang-type determination, this approach may result in several biases, such as social desirability [20], measurement [21], and recall [22] biases. Measurement bias is particularly problematic when using self-administered questionnaires for Sasang determination because the language used in the questionnaire can be confusing and difficult to answer accurately, making it difficult to distinguish different Sasang constitutional types. A machine-learning approach has recently been introduced to improve the accuracy of Sasang constitution type identification; however, such efforts remain limited in the Sasang constitution medicine literature [23].

Moreover, previous studies classified the respondents into one of the Sasang constitutional types, although some Sasang constitutional types may share similar pathophysiological or psychological traits [24,25]. Hence, this study aimed to discover the Sasang determination questionnaire items and key constructs that produce significantly different responses for each Sasang constitution. For this purpose, this study tested a wide range of measurement items widely implemented in previous studies and fields and implemented a feature selection algorithm to discover significant constructs and questionnaire items. The similarities and differences across the Sasang constitutional types were identified using various clustering approaches. 

The rest of this paper is structured as follows. The second section describes the related works. The third section outlines materials and methods used in the study. The fourth section presents the empirical findings, and the fifth section includes a discussion. The final section presents conclusions.

## 2. Related Works

Sasang constitutional medicine has a long history and serves as a clinical classification system that categorizes persons into one of four distinct groups, each of which is meant to reflect the prevalent characteristics of each group [26]. The Sasang constitutional theory was developed in order to distinguish individual differences in receptivity to a particular treatment and the effectiveness of medicine [8]. Sasang constitutional theory demonstrates the principles of contemporary preventive medicine, which promote preventive health care through healthy lifestyles and individualized treatment [27]. 

With the growing popularity and familiarity of Sasang constitutional medicine among Koreans, various fields other than medicine have paid attention to its utility. In a previous study with 839 participants, for example, 88.4% of the surveyed reported that it is crucial to choose food ingredients that are appropriate for their Sasang typology for disease prevention and treatment [28]. Moreover, Sasang constitutional medicine has sparked interest in psychology [29], media [30], fashion [31,32], exercise [33,34], animation [35], and wine recommendation systems [36], demonstrating its interdisciplinary potential in the era of personalized medicine. To expand international use of Sasang typology, Sasang constitutional medicine has been applied to people from countries other than Korea, including Chinese [37], Americans [38], and Vietnamese [39]. 

Nevertheless, Sasang constitutional medicine lacks universally accepted quantitative criteria for identifying the Sasang constitutional typology; hence, the diagnostic results have been deemed subjective and confusing [40]. To address this issue, extensive efforts have been made to develop reliable and accurate Sasang constitutional identification tools. For Sasang constitutional typology identification, various sub-constructs are considered, including pathophysiological characteristics [41,42,43,44,45,46], personality characteristics [43,44] body shape appearance [44], digestive functions and eating habits [46]. As a result, Sasang typology investigates not only clinical and physiological traits, but also the broad ranges of various sub-constructs in order to comprehend both the body and the mind [47,48]

To further increase the usability of Sasang constitutional medicine, previous studies argue that the Sasang constitutional identification tool needs improvements in reliability [49,50]. For example, over the course of a year, Bae, Kim, Go, Park, Lee and Lee [50] administered the same Sasang constitutional diagnostic questionnaire to university students twice. Despite no significant physical changes (i.e., BMI results), nearly one-third of the participants had different Sasang constitutional type results from the previous year, indicating the need for a more reliable questionnaire. Similarly. Lee, Yim and Kim [49] conducted a test-retest reliability test with SCAT over four weeks and found some sub-constructs of the tool had low correlation coefficients. Therefore, developing a reliable, user-friendly, and integrative tool for the Sasang-type determination is necessary. 

## 3. Materials and Methods

### 3.1. Study Participants and Recruitment

This study was conducted with healthy adult men and women aged >20 years who lived in metropolitan areas, including Seoul, Gyeonggi-do, and Incheon, Republic of Korea, from 1 November 2018 to 30 June 2019. Participants were recruited through recruitment brochures, banner advertisements, and internet posts that detailed the research plan and procedures. All the individuals voluntarily participated in this study, and 419 finished the survey and constitutional type identification. After removing 13 incomplete survey results, 406 surveys were used for data analysis. This study was approved by the Institutional Review Board of Gachon University (IRB No. 044396-201808-HR-171-01). 

### 3.2. Identification of Sasang Constitutional Types

The Sasang constitution was identified by Sasang experts in the private sector. Following previous studies [51,52], various methods, including pendulum and O-ring, were used for Sasang constitution identification. Aroma oil and color cards (Tae-Yang, green; So-Yang, blue; Tae-Eum, yellow; and So-Eum, orange) were used as reference materials to ensure the accuracy of the O-ring test. DoTERRA (DoTERRA Korea, Seoul) was used as a reference material for aroma oil. Four different scents (lemon, grapefruit, peppermint, and rosemary) were used for Tae-Yang, So-Yang, Tae-Eum, and So-Eum, respectively. We found that respondents classified as Tae-Yang types were rare, which is consistent with earlier studies [23,40]. Tae-Yang types were excluded from the data analysis because it was crucial to have a sufficient number of respondents in each Sasang constitution for data analysis.

### 3.3. Measurement Items

The questionnaire included 225 questions based on the SCAT 2.0 questionnaire by the Korea Institute of Oriental Medicine, ideological medicine books [53], and field experiences. Among the 225 questions, 133 were about pathophysiological characteristics [41,42,43,44,45,46], and 92 were related to personality characteristics [43,44]. The questionnaire items include physiological symptoms about cold and heat [41] (e.g., My hands are cold), body shape appearance [44] (e.g., I have a petite and slender physique), digestive functions and eating habits (e.g., I usually eat slowly) [46]. Some example items regarding personality characteristics are “I tend to be inactive”, “I am sociable”, “I enjoy speaking in public”. All questions were answered with ‘yes (coded as 1) or no (coded as 0). This study included several similar questionnaire items on the same attribute to identify the types of questions that reduce confusion among the respondents and confirm the internal consistency of the survey responses on the same attribute.

### 3.4. Data Analysis

#### 3.4.1. Feature Selection Algorithm

One of the objectives of this research was to select essential items that determine differences across Sasang constitutional types from a large collection of comprehensive measurement items collected from various sources. To this end, a feature selection algorithm was applied to identify key items that were highly relevant to the target variable (Sasang constitutional types). Because our target variable (i.e., Sasang constitutional types) was a categorical variable, we implemented a chi-squared feature selection. The Python library “Scikit-learn” (version 0.21.2) was used in the Jupyter notebook for feature selection (Figure 1). The 227 collected items served as sample vectors for the analysis, with three Sasang constitutional types serving as the target vectors. 

The Scikit-Learn feature selection produced χ^2^ statistics for each item, with a higher χ^2^ of sample vectors indicating greater relevance to the target variable. Therefore, the χ^2^ scores assigned to each item were used to assess the degree of relevance of the 227 measurements to Sasang constitutional types. The item selection cut-off value was established at 4.605 (df = 2, 90% confidence interval), leaving 47 items with high scores for further data analysis (Figure 1). 

#### 3.4.2. Clustering Analysis

Clustering is a widely used unsupervised machine-learning method for grouping unlabeled data. A hierarchical clustering approach was adopted to uncover the clustering of the measurement items and organization of these clusters. The clustering algorithm generates several clusters that are internally and externally distinct. Distance measures were used to assess the similarity of items inside a cluster and dissimilarity to those in other clusters. Based on graph theory, the relationships between the items were represented by a hierarchical cluster dendrogram [54,55]. According to graph theory, graphs are architectures consisting of a collection of nodes and edges and that describe data relationships. The hierarchical clustering technique aims to produce a network structure (dendrogram) by linking clusters based on similarity and grouping criteria. In other words, the hierarchical clustering technique initially connects the two most similar clusters and continues the process of constructing larger clusters.

Hierarchical clustering analysis was performed using the R package “factoextra” (version 1.0.6), which allowed multivariate data processing and visualization. Prior to clustering analysis, data pre-processing was performed. Individual responses for the 47 items were totaled to compute the average score for each measurement item. Hence, the variables that were originally binary (i.e., recorded as 0 or 1) were converted into continuous numerical variables. The “hclust” function in “factoextra” was used to perform hierarchical clustering with six clusters. The agglomeration method was set up as “ward.D2,” and the “euclidean” dissimilarity metric was adopted. Figure 2 illustrates the procedure of the proposed method. 

## 4. Results

### 4.1. Clustering in Sasang Questionnaire Items

Table 1 and Figure 3 present the results of the hierarchical clustering analysis. Table 1 shows the 47 selected items of feature selection and the Sasang typology constructs associated with each item. In addition, Table 1 displays the average scores for each item according to the Sasang constitutional type and the rank of the three Sasang constitutional types. The dendrogram in Figure 3 depicts the similarities and dissimilarities of the 47 items. Similar items are located close to each other and linked to form a small group, and the process of joining the subgroups is repeated to build clusters. The height indicates the degree of similarity between two items. The shorter dendrogram height indicates that items with shorter heights are closer than others with longer dendrogram heights. For instance, the dendrogram height that links items 27 and 44 is the shortest, suggesting that these two items are the most similar. 

As a result of the hierarchical clustering analysis, we found that the first cluster consisted of three items that were mostly concerned with personality. Specifically, two of the three items in the first cluster (Items 9 and 197) were related to personality traits, while the other was related to pathophysiology (Item 200). The dendrogram illustrates that two personality items (items 9 and 197) about being meticulous and timid were more closely related to each other than the other pathophysiological items about being sweaty in the same cluster. The So-Eum type outperformed the Tae-Eum and So-Yang types on all items in the first cluster, implying that introverted personality traits and less sweating were more prominent in the So-Eum type than that in the other types. This result is consistent with a previous study, which reported that the So-Eum type was less extroverted than other types [56]. 

The second cluster, with 16 items, had two primary branches. The majority of items in the left branch were about the personality traits of being active and adventurous. The items in the right branch were related to pathophysiological symptoms, such as perspiration, diet, weight, skin condition, and preferred temperature. The pathophysiological symptoms of Cluster 2 were sweating, toasty body, excessive eating, and rapid weight gain. The So-Yang and Tae-Eum types had high scores for the items in the second cluster, indicating that these two types shared similar traits in terms of personality and pathophysiology. More specifically, the So-Yang type had the highest scores on the personality items in the second cluster, followed by the Tae-Eum and So-Yang types, whereas the Tae-Eum type had the highest scores on the pathophysiology items in the second cluster, followed by So-Yang and So-Eum.

The third cluster largely consisted of items about friendliness, sociability, and optimism. All these traits were most noticeable in So-Yang, followed by Tae-Eum and So-Eum. The fourth cluster mostly comprised items on pathophysiology, and Tae-Eum had the highest scores on these items. Items in the third and fourth clusters were comparable to those in the second cluster. Although the items in the second cluster had high scores for both So-Yang and Tae-Eum, the items in the third and fourth clusters were most prevalent for So-Yang and Tae-Eum, respectively.

The items in the fifth cluster represented the pathophysiological characteristics of the So-Eum type, which differed from the symptoms illustrated in the second, third, and fourth clusters. Many respondents with the So-Eum type indicated that their hands, feet, and stomach felt cold, they did not sweat much, and consumed a small amount of food. So-Eum had the highest scores for the items of the first and fifth clusters. However, So-Yang had the second highest scores in most items of the fifth cluster, while the Tae-Eum type had the second highest scores in the first cluster, followed by So-Eum. 

Finally, the sixth cluster consisted of personality and pathophysiological traits of the So-Eum and Tae-Eum types. Although the second cluster indicated personality and pathophysiological symptoms relevant to both So-Yang and Tae-Eum, the sixth cluster suggested that So-Eum and Tae-Eum shared similar characteristics.

### 4.2. Findings of Multiple Correspondence Analysis 

Multiple correspondence analysis was performed to summarize the cluster analysis results and understand the relationships between the clusters (Figure 4). The first and second dimensions accounted for 57.8% and 29.6% of the data variability, respectively. In total, the two dimensions accounted for 87.4% of the data variability.

Dimension 1 on the y-axis divided the clusters that were noticeably applicable to a particular Sasang constitution from those that are commonly applicable to multiple Sasang constitutional types. Clusters salient to a single Sasang constitution (i.e., the first, third, and fourth clusters) were placed on the negative side of dimension 1 (left side of the graph). For instance, all the items in Cluster 1 had the highest scores for the So-Eum type. Similarly, all the items in cluster 3 had the highest scores in So-Yang, and all items in Cluster 4 had the highest scores in the Tae-Eum type. Clusters prominent in the two Sasang constitutional types (i.e., the second and sixth clusters) were located on the positive side of dimension 1 (right side of the graph). For instance, Cluster 2 was skewed to the right side of the graph, and both the So-Yang and Tae-Eum types had high scores in the items of Cluster 2. Similarly, both the So-Eum and Tae-Eum types had high scores in the items of Cluster 6.

Dimension 2 on the X-axis divided the clusters based on whether they were mainly relevant to the So-Eum type or the other two Sasang constitutional types. The first and fifth clusters were placed on the negative side (bottom side of the graph), and these clusters demonstrated So-Eum-type traits. The second, third, fourth, and sixth clusters, which represent the So-Yang and Tae-Eum characteristics, were placed on the positive side of dimension 2 (upper side of the graph). This result implies that So-Yang and Tae-Eum types tend to share more similar pathophysiological and personality traits than the So-Eum type. 

### 4.3. Distances among the Clusters and Measurement Items

A heatmap analysis was conducted to confirm the relationships among the measurement items according to Sasang constitutional types and clusters (Figure 5). The dark red block in Figure 5 indicates that the corresponding item has a strongly positive relationship with each Sasang constitution, whereas the dark blue color indicates a strong negative relationship with the Sasang constitution. The solid bright blue line in the heatmap indicates the distance between each item and the Sasang constitution. More specifically, the dark red colors of the heatmap and the blueline of the So-Yang type indicate that this type is closely related to measurement items 168 (i.e., I am sociable) and 91 (i.e., even when I first meet someone, I quickly get friendly with them) in Cluster 3. Similarly, the Tae-Eum type had a close relationship with item 156 (i.e., My hands are toasty) in Cluster 4 and 168 (i.e., I am sociable) in Cluster 3. Finally, So-Eum type had a close relationship with items 9 (i.e., because I am meticulous, I do not make mistakes) and 200 (i.e., I normally sweat less) in Cluster 1.

In addition, the relationships among the clusters can be identified based on the dendrogram at the top of the heatmap. In general, the So-Yang and Tae-Eum types shared more similar attributes than the So-Eum type, indicating that the So-Eum type tended to have more unique pathophysiological and personality traits than the So-Yang and Tae-Eum types. Still, So-Eum had similar traits to So-Yang and Tae-Eum types for items that had similar heatmap color shades.

## 5. Discussion

Sasang constitutional medicine has a long history with an emphasis on disease prevention and personalized treatment, which is consistent with contemporary preventive medicine [27]. Sasang constitutional medicine not only considers physical and clinical characteristics for personalized medicine, but also personality traits and various behavioral characteristics (e.g., eating attitude and dietary behaviors) [57,58,59,60]. Consequently, the Sasang constitution typology can serve as a guide for healthy lifestyles and the quest for mutual respect and comprehension of the four distinct constitutions [59]. Although Sasang constitutional medicine has gained appeal among the general public, the validity and reliability of Sasang constitution determination tools remains a source of concern. In addition, previous studies concerned with the development of a reliable Sasang constitution typology identification tool tended to place an excessive amount of emphasis on classifying the constitutional groups based on the origin of the Sasang constitutional medicine, *Donguisusebowon*, which was published at the end of the nineteenth century [57]. Consequently, it is necessary to evaluate the efficacy of numerous sub-constructs of the Sasang constitution typology using cutting-edge approaches.

In order to address this issue, this study compiled previously utilized sub-constructs and questionnaire items, as well as recently validated/suggested sub-constructs and questionnaire items, in order to determine the most effective sub-constructs and questionnaire items for distinguishing Sasang constitution typologies. In addition, different rephrased questionnaire items were evaluated to determine which wording or phrases are more effective for identifying Sasang constitution type to improve the usability of the Sasang typology identification in many commercial areas. Most importantly, this study employed clustering methods to discover pathophysiological and personality traits that are specific to each Sasang type, as well as overlapping traits that are applicable to several Sasang types. In summary, this study tested and analyzed existing Sasang constitution questionnaire items to provide future resources for developing a robust Sasang analysis tool. This study suggests meaningful commercial and academic implications by adopting various machine learning approaches that have rarely been implemented in Sasang constitutional medicine studies.

First, this study conducted feature selection to determine which attributes produced different responses across Sasang constitutional types. Previous studies attempted to select key measurement items from a large set of measurement items used in prior studies and in the field to reduce survey respondents’ fatigue resulting from answering a large number of questions and thereby enhance the outcomes of Sasang constitution identification [14,61]. This study proposes a new strategy to discover relevant attributes for Sasang constitution identification and improve future machine learning outcomes by implementing a machine-learning algorithm called feature selection. As a result, this study discovered pathophysiological symptoms and personality are both important constructs to identify the Sasang constitution types. For instance, questions regarding sweat and temperature were found to be effective to distinguish different Sasang types in addition to questions regarding eating habits and body shapes. Furthermore, many questionnaire items regarding personality were found to be effective for Sasang constitution identification. 

Second, the results of the hierarchical clustering analysis revealed that some attributes are salient to one Sasang constitution, whereas others may be applicable to several types. The findings imply that certain attributes are unique characteristics of a particular Sasang constitution, and some attributes can be found in various Sasang constitutional types. Previous research suggests that Tae-Yang and So-Yang typologies, as well as So-Yang and So-Eum typologies, share similar biological traits and, thus, exhibit comparable behavioral and psychological traits [13,62]. This study not only verifies the findings of prior research, but also illustrates the overlapping constructs and prominent characteristics of several typologies. Our findings, therefore, can be useful for determining the attributes that represent the distinct characteristics of each Sasang constitution to develop a time efficient and effective questionnaire, especially when the Sasang constitution questionnaire is used as a supplementary tool along with other methods. However, including measurement items applicable to several types can be useful in developing a comprehensive Sasang constitution questionnaire. In this case, the results should be interpreted based on understanding the relationship between each question and Sasang constitution, rather than simple summary methods, such as the sum or average of all item results. 

Finally, multiple correspondence results and a heatmap analysis revealed that the Tae-Eum and So-Yang types tend to share more similar pathophysiological and psychological traits than the So-Eum type. In addition, this study suggests specific areas in which Sasang constitutional types have a significant degree of similarity and dissimilarity.

This study is useful for the development of more robust Sasang constitution questionnaires, so that Sasang constitution medicine can be applied in various commercial fields. By including multiple rephrased questionnaire items about the same trait, this study examined which wording is more clearly comprehensible to survey participants and is more beneficial in identifying the Sasang Constitution. 

This study suggests that a person identified with a specific Sasang constitution has some pathophysiological and psychological characteristics of other Sasang constitutional types. Instead of classifying certain attributes into the characteristics of a particular Sasang constitution, this study identifies comparable and unique attributes among Sasang constitutional types. These discoveries have the potential to improve future Sasang identification accuracy, particularly in individuals who exhibit pathophysiological and psychological traits of various Sasang constitutional types.

## 6. Conclusions

This study employs an exploratory approach to uncover the important constructs and questionnaire items that distinguish different Sasang constitutional types and to discover the similarities and differences among the Sasang constitutional types. To achieve these aims, this study implemented various novel approaches such as feature selection, clustering analysis, and multiple correspondence analysis. Hence, the study provides useful insights for the future development and deployment of Sasang constitution questionnaires to reduce survey respondent confusion and improve Sasang determination accuracy. Moreover, this study introduced new methodologies into this academic field.

Despite the contributions of this study, further improvements should be aimed for in future study. First, this study excluded the Tae-Yang type because only a small number of survey participants were identified belonging to this type. Future studies should include more respondents of the Tae-Yang type to identify the relationships between the four different Sasang constitutional types. Second, this study was based on Sasang identification results from one expert in the private sector. Future studies may consider identifying Sasang constitutional types based on the collaboration of several Sasang experts in the private sector and Korean medicine doctors. As suggested by previous study [36], digital technology (e.g., face recognition application) can be implemented to improve the accuracy of Sasang constitution identification results. Finally, this study did not consider covariates (e.g., age and sex) that may influence the relationships between attributes and Sasang constitutional types. Future studies should be conducted to explore the influence of age or gender on the Sasang constitution results.

## Figures and Tables

**Figure 1 ijerph-19-11820-f001:**
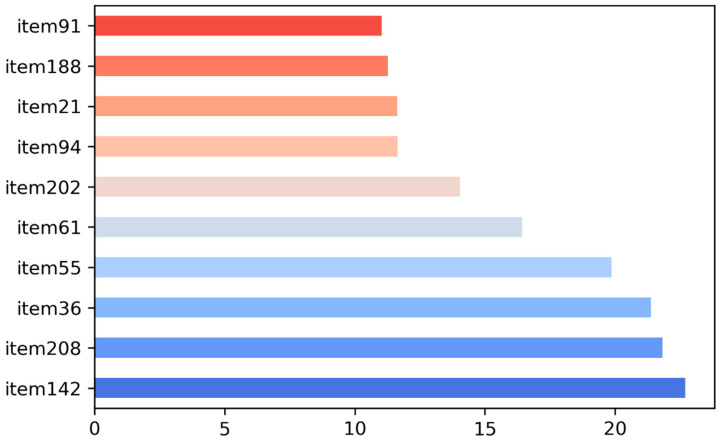
Top ten measurement items sorted by feature scores.

**Figure 2 ijerph-19-11820-f002:**
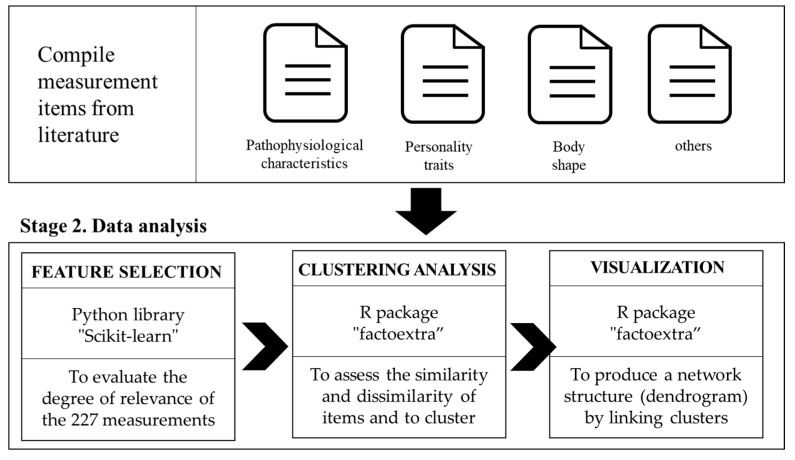
The procedure of the proposed method.

**Figure 3 ijerph-19-11820-f003:**
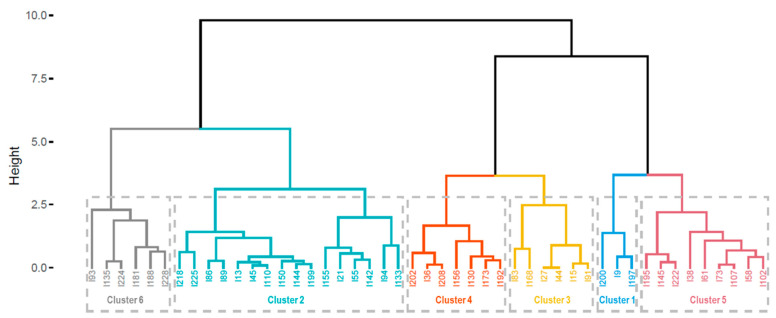
Hierarchical cluster dendrogram.

**Figure 4 ijerph-19-11820-f004:**
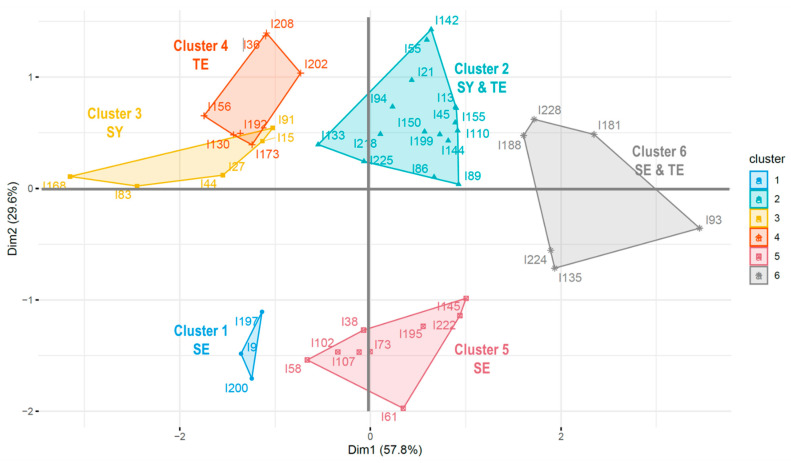
Multiple correspondence analysis results.

**Figure 5 ijerph-19-11820-f005:**
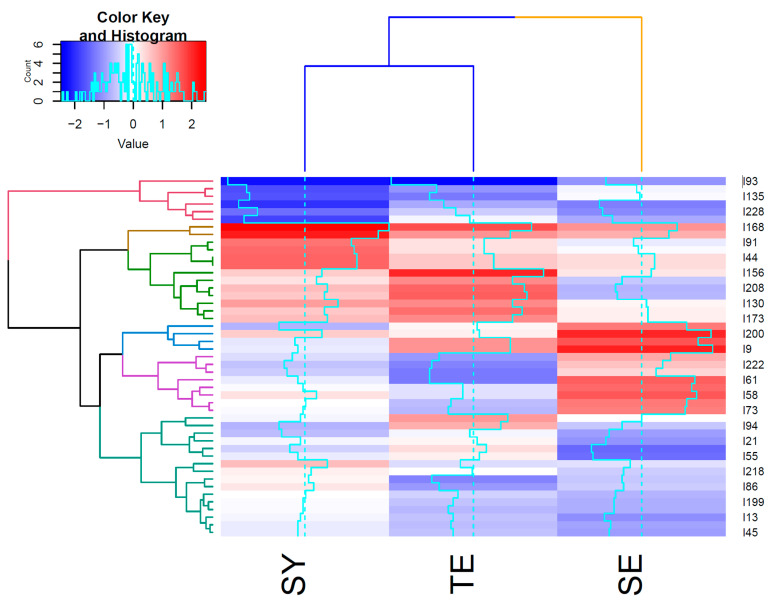
Distances among clusters and measurement items.

**Table 1 ijerph-19-11820-t001:** Summary of hierarchical clustering.

Cluster	Item Number	Items	Constructs	SE	SY	TE	Order
1	I9	Because I am meticulous, I do not make mistakes	Personality	0.65	0.39	0.56	SE > TE > SY
	I197	When I’m offended, my hurtful feelings stay a long time	Personality	0.59	0.38	0.56	SE > TE > SY
	I200	I normally sweat less	Pathophysiology	0.64	0.49	0.44	SE > TE > SY
2	I45	My action seems to be ahead of my thoughts	Personality	0.22	0.39	0.34	SY > TE > SE
	I110	I tend to lighten up the mood in unfamiliar situations	Personality	0.23	0.39	0.33	SY > TE > SE
	I144	I am sloppy and prone to making mistakes	Personality	0.24	0.41	0.32	SY > TE > SE
	I199	I enjoy speaking in public	Personality	0.24	0.41	0.34	SY > TE > SE
	I150	Because I start multiple projects at once, I frequently fail to finish them all	Personality	0.25	0.42	0.36	SY > TE > SE
	I89	I like to get up early	Personality	0.27	0.43	0.25	SY > SE > TE
	I218	I have a tendency to begin with ambition but fail to wrap up successfully	Personality	0.28	0.44	0.41	SY > TE > SE
	I225	I am more interested in new experiences than in things I should do	Personality	0.31	0.51	0.36	SY > TE > SE
	I199	I enjoy speaking in public	Personality	0.24	0.41	0.34	SY > TE > SE
	I86	I frequently experience indigestion	Pathophysiology	0.28	0.46	0.28	SY > TE = SE
	I155	I am sweaty	Pathophysiology	0.22	0.32	0.40	TE > SE > SY
	I21	I normally sweat a lot	Pathophysiology	0.21	0.40	0.43	TE > SE > SY
	I55	I normally eat a lot throughout the day	Pathophysiology	0.15	0.39	0.44	TE > SE > SY
	I142	Even if I eat a small amount of food, I tend to gain weight quickly	Pathophysiology	0.15	0.36	0.47	TE > SE > SY
	I94	My skin is thick and tight	Pathophysiology	0.27	0.31	0.52	TE > SE > SY
	I133	My feet are toasty	Pathophysiology	0.35	0.40	0.55	TE > SE > SY
3	I15	The upper body (shoulder) is larger than the lower body (hip)	Body shape	0.34	0.62	0.46	SY > TE > SE
	I168	I am sociable	Personality	0.5	0.77	0.64	SY > TE > SE
	I83	I am friendly	Personality	0.47	0.72	0.56	SY > TE > SE
	I27	I have a habit of becoming involved in other people’s business as if it were my own	Personality	0.41	0.63	0.5	SY > TE > SE
	I44	I am optimistic and driven	Personality	0.41	0.63	0.5	SY > TE > SE
	I91	Even when I first meet someone, I quickly get friendly with them	Personality	0.32	0.61	0.46	SY > TE > SE
4	I156	My hands are toasty	Pathophysiology	0.39	0.49	0.69	TE > SY > SE
	I36	I tend to gain weight easily	Pathophysiology	0.25	0.50	0.63	TE > SY > SE
	I208	I have an excessively good appetite	Pathophysiology	0.25	0.51	0.62	TE > SY > SE
	I192	I usually sweat a lot when it is hot	Pathophysiology	0.38	0.50	0.61	TE > SY > SE
	I130	I typically drink cold water	Pathophysiology	0.37	0.56	0.57	TE > SY > SE
	I173	I usually sweat a lot and feel revitalized afterwards	Pathophysiology	0.38	0.51	0.57	TE > SY > SE
	I202	I have a good and powerful physique	Body shape	0.27	0.47	0.57	TE > SY > SE
5	I61	I have a petite and slender physique	Body shape	0.57	0.39	0.24	SE > SY > TE
	I38	The lower body (hip) is larger than the upper body (shoulder)	Body shape	0.54	0.31	0.43	SE > TE > SY
	I58	I do not sweat much	Pathophysiology	0.58	0.47	0.37	SE > SY > TE
	I102	My hands are cold	Pathophysiology	0.56	0.41	0.37	SE > SY > TE
	I107	I typically eat a small amount of food per day	Pathophysiology	0.54	0.42	0.33	SE > SY > TE
	I73	My skin is thin and sensitive	Pathophysiology	0.53	0.41	0.32	SE > SY > TE
	I195	My hands and feet are cold	Pathophysiology	0.47	0.37	0.28	SE > SY > TE
	I222	My stomach feels quite cold	Pathophysiology	0.44	0.33	0.25	SE > SY > TE
	I145	I usually eat slowly	Pathophysiology	0.41	0.34	0.24	SE > SY > TE
6	I135	It is hard for me to get along with new people because I am not outgoing	Personality	0.35	0.19	0.24	SE > TE > SY
	I224	I tend to be inactive	Personality	0.33	0.18	0.27	SE > TE > SY
	I228	I am unconcerned about what people think of me	Personality	0.19	0.22	0.35	TE > SY > SE
	I181	I am slow to act and make decisions	Personality	0.18	0.14	0.30	TE > SE > SY
	I188	I tend to sweat a lot when I’m not feeling well	Pathophysiology	0.23	0.17	0.40	TE > SE > SY
	I93	I have a poor appetite	Pathophysiology	0.21	0.10	0.09	SE > SY > TE

Note. SE: So-Eum, SY: So-Yang, TE: Tae-Eum, organized according to the cluster numbers and order of the items in the dendrogram (Figure 3).

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
