# Peer review of "Machine Learning Applications for the Development of a Questionnaire to Identify Sasang Constitution Typology"

_ijerph, 2022, doi:10.3390/ijerph191811820_

Round 1
Reviewer 1 Report
This study takes an exploratory approach to uncover the essential constructs and questionnaire items that distinguish different Sasang constitutional types and to discover similarities and differences among the Sasang constitutional types. The research is fascinating. After reading the manuscript, here are comments:
- In this research, self-reported surveys are widely used for the Sasang-type determination. This approach may result in several biases, such as social desirability, measurement, and recall. Measurement bias is particularly problematic when using self-administered questionnaires for Sasang determination because the language used in the questionnaire can be confusing and challenging to answer accurately, making it difficult to distinguish different Sasang constitutional types. The novelty of this study is not apparent. How the approach proposed by the authors differs from other similar approaches? What is your difference from the previous works and innovation? Clearly state the novelty of this study at the end of the introduction section.
- The contribution of this work is still limited domain. Sasang constitutional medicine has a long history with an emphasis on disease prevention and personalized treatment, consistent with contemporary preventive medicine. How is this research vital in modern medical treatment?
- The scope of this research is not complete. In line 307, this study can be helpful for the development of more robust Sasang constitution questionnaires so that Sasang constitution medicine can be applied in various industrial fields. By including multiple rephrased questionnaire items about the same trait, this study examined which wording is more clearly comprehensible to survey participants and is more beneficial in identifying the Sasang Constitution. Why should the authors perform all of this in this research version? Please clarify.
- The authors failed to present the related works sufficiently (consequence from comments 2.).
- In line 291, the results of the hierarchical clustering analysis revealed that some attributes are salient to one Sasang constitution, whereas others may apply to several types. How can the authors conclude with this point? Please give the reasons.
- How many did participants be in this study? Please identify.
- Figure 1 is not necessary.
- Please clarify Figure 1 in "The dendrogram in Figure 1 depicts the similarities and dissimilarities....." of line 162.
- Table 1 is needed to be rearrangement.
Author Response
|
Comments from reviewers |
Responses |
|
This study takes an exploratory approach to uncover the essential constructs and questionnaire items that distinguish different Sasang constitutional types and to discover similarities and differences among the Sasang constitutional types. The research is fascinating. After reading the manuscript, here are comments: |
Thank you for your time and work in reviewing our paper. We made every effort to respond to all of your comments thoroughly. Please see below for our responses to your comments. Thanks. |
|
1. In this research, self-reported surveys are widely used for the Sasang-type determination. This approach may result in several biases, such as social desirability, measurement, and recall. Measurement bias is particularly problematic when using self-administered questionnaires for Sasang determination because the language used in the questionnaire can be confusing and challenging to answer accurately, making it difficult to distinguish different Sasang constitutional types. The novelty of this study is not apparent. How the approach proposed by the authors differs from other similar approaches? What is your difference from the previous works and innovation? Clearly state the novelty of this study at the end of the introduction section. |
Thank you for your comment. To emphasize the research gap and the value of this study, we revised Introduction section. Please see the highlighted part. Thank you.
[Lines 76-102] In addition, previous studies reported that Sasang constitutional identification tool needs to improve the reliability [33,34]. For example, over the course of a year, Bae, Kim, Go, Park, Lee and Lee [34] administered the same Sasang constitutional diagnostic questionnaire to university students twice. Despite no significant physical changes (i.e., BMI result), nearly one-third of the participants had different Sasang constitutional type results from the previous year, indicating the need for a more reliable questionnaire. Similarly. Lee, Yim and Kim [33] conducted test-retest reliability test with SCAT over the four weeks and found some sub-constructs of the tool had low correlation coefficient. Therefore, developing a reliable, user-friendly, and integrative tool for the Sasang-type determination is necessary. Although self-reported surveys are widely used for the Sasang-type determination, this approach may result in several biases, such as social desirability [35], measurement [36], and recall [37] biases. Measurement bias is particularly problematic when using self-administered questionnaires for Sasang determination because the language used in the questionnaire can be confusing and difficult to answer accurately, making it difficult to distinguish different Sasang constitutional types. A machine-learning approach has recently been introduced to improve the accuracy of Sasang constitution type identification; however, such effort remains limited in the Sasang constitution medicine literature [38]. Moreover, previous studies classified the respondents into one of the Sasang constitutional types, although some Sasang constitutional types may share similar pathophysiological or psychological traits [39,40]. Hence, this study aimed to discover the Sasang determination questionnaire items and key constructs that produce significantly different responses for each Sasang constitution. For this purpose, this study tested a wide range of measurement items widely implemented in previous studies and fields and implemented a feature selection algorithm to discover significant constructs and questionnaire items. The similarities and differences across the Sasang constitutional types were identified using various clustering approaches.
|
|
2. The contribution of this work is still limited domain. Sasang constitutional medicine has a long history with an emphasis on disease prevention and personalized treatment, consistent with contemporary preventive medicine. How is this research vital in modern medical treatment? |
To address this comment, we provided more detailed information about how this study fills the research gap in Discussion section.
[Lines 295-323] Sasang constitutional medicine has a long history with an emphasis on disease prevention and personalized treatment, which is consistent with contemporary preventive medicine [10]. Sasang constitutional medicine not only considers physical and clinical characteristics for personalized medicine, but also personality traits and various behavioral characteristics (e.g., eating attitude and dietary behaviors) [54-57]. Consequently, the Sasang constitution typology can serve as a guide for healthy lifestyles and the quest of mutual respect and comprehension of the four distinct constitutions [56]. Although Sasang constitutional medicine has gained appeal among the general public, the validity and reliability of Sasang constitution determination tools remains a source of concern. In addition, previous studies concerned with the development of a reliable Sasang constitution typology identification tool tended to place an excessive amount of emphasis on classifying the constitutional groups based on the origin of the Sasang constitutional medicine, Donguisusebowon, which was published at the end of the nineteenth century [54]. Consequently, it is necessary to evaluate the efficacy of numerous sub-constructs of the Sasang constitution typology using cutting-edge approaches. In order to address this issue, this study compiled previously utilized sub-constructs and questionnaire items, as well as recently validated/suggested sub-constructs and questionnaire items, in order to determine the most effective sub-constructs and questionnaire items for distinguishing Sasang constitution typologies. In addition, different rephrased questionnaire items were evaluated to determine which wording or phrases are more effective for identifying Sasang constitution type to improve the usability of the Sasang typology identification in many industries. Most importantly, this study employed clustering methods to discover pathophysiological and personality traits that are specific to each Sasang type, as well as overlapping traits that are applicable to several Sasang types. In summary, this study tested and analyzed existing Sasang constitution questionnaire items to provide future resources for developing a robust Sasang analysis tool. This study suggests meaningful industrial and academic implications by adopting various machine learning approaches that have rarely been implemented in Sasang constitutional medicine studies.
|
|
3. The scope of this research is not complete. In line 307, this study can be helpful for the development of more robust Sasang constitution questionnaires so that Sasang constitution medicine can be applied in various industrial fields. By including multiple rephrased questionnaire items about the same trait, this study examined which wording is more clearly comprehensible to survey participants and is more beneficial in identifying the Sasang Constitution. Why should the authors perform all of this in this research version? Please clarify. |
Thank you for bringing up an excellent suggestion to enhance the quality of this manuscript. Your feedback has been addressed in the Discussion section. Thank you very much.
[Lines 310-323] In order to address this issue, this study compiled previously utilized sub-constructs and questionnaire items, as well as recently validated/suggested sub-constructs and questionnaire items, in order to determine the most effective sub-constructs and questionnaire items for distinguishing Sasang constitution typologies. In addition, different rephrased questionnaire items were evaluated to determine which wording or phrases are more effective for identifying Sasang constitution type to improve the usability of the Sasang typology identification in many industries. Most importantly, this study employed clustering methods to discover pathophysiological and personality traits that are specific to each Sasang type, as well as overlapping traits that are applicable to several Sasang types. In summary, this study tested and analyzed existing Sasang constitution questionnaire items to provide future resources for developing a robust Sasang analysis tool. This study suggests meaningful industrial and academic implications by adopting various machine learning approaches that have rarely been implemented in Sasang constitutional medicine studies.
|
|
4. The authors failed to present the related works sufficiently (consequence from comments 2.). |
In relation to comment 2, please see the revised version of our manuscript. Thanks.
[Lines 76-93] In addition, previous studies reported that Sasang constitutional identification tool needs to improve the reliability [33,34]. For example, over the course of a year, Bae, Kim, Go, Park, Lee and Lee [34] administered the same Sasang constitutional diagnostic questionnaire to university students twice. Despite no significant physical changes (i.e., BMI result), nearly one-third of the participants had different Sasang constitutional type results from the previous year, indicating the need for a more reliable questionnaire. Similarly. Lee, Yim and Kim [33] conducted test-retest reliability test with SCAT over the four weeks and found some sub-constructs of the tool had low correlation coefficient. Therefore, developing a reliable, user-friendly, and integrative tool for the Sasang-type determination is necessary. Although self-reported surveys are widely used for the Sasang-type determination, this approach may result in several biases, such as social desirability [35], measurement [36], and recall [37] biases. Measurement bias is particularly problematic when using self-administered questionnaires for Sasang determination because the language used in the questionnaire can be confusing and difficult to answer accurately, making it difficult to distinguish different Sasang constitutional types. A machine-learning approach has recently been introduced to improve the accuracy of Sasang constitution type identification; however, such effort remains limited in the Sasang constitution medicine literature [38].
|
|
5. In line 291, the results of the hierarchical clustering analysis revealed that some attributes are salient to one Sasang constitution, whereas others may apply to several types. How can the authors conclude with this point? Please give the reasons. |
In Discussion section, we addressed how our findings can be understood in relation to previous studies.
[Lines 337-345] Second, the results of the hierarchical clustering analysis revealed that some attributes are salient to one Sasang constitution, whereas others may be applicable to several types. The findings imply that certain attributes are unique characteristics of a particular Sasang constitution, and some attributes can be found in various Sasang constitutional types. Previous research suggests that Tae-Yang and So-Yang typologies, as well as So-Yang and So-Eum typologies, share similar biological traits and, thus, exhibit comparable behavioral and psychological traits [26,59]. This study not only verifies the findings of prior research, but also illustrates the overlapping constructs and prominent characteristics of several typologies. |
|
6. How many did participants be in this study? Please identify. |
[Lines 109-111] All the individuals voluntarily participated in this study, and 419 finished the survey and constitutional type identification. After removing 13 incomplete survey results, 406 surveys were used for data analysis. |
|
7. Figure 1 is not necessary. |
We removed Figure 1. |
|
8. Please clarify Figure 1 in "The dendrogram in Figure 1 depicts the similarities and dissimilarities....." of line 162. |
Thank you for pointing out this, and we fixed Figure number. We also explained the dendrogram and hierarchical clustering analysis both in Result and Method.
[Lines 184-190] The dendrogram in Figure 2 depicts the similarities and dissimilarities of the 47 items. Similar items are located close to each other to be linked and to form a small group, and the process of joining the subgroups is repeated to build clusters. The height indicates the degree of similarity between two items, as the shorter dendrogram height indicates that items with shorter heights are closer than others with longer dendrogram heights. For instance, the dendrogram height that links items 27 and 44 is the shortest, suggesting that these two items are the most similar.
[Lines 160-169] The clustering algorithm generates several clusters that are internally and externally distinct. Distance measures were used to assess the similarity of items inside a cluster and dissimilarity to those in other clusters. Based on graph theory, the relationships between the items were represented by the hierarchical cluster dendrogram [51,52]. According to graph theory, graphs are architectures consisting of a collection of nodes and edges and that describe data relationships. Hierarchical clustering technique aims to produce a network structure (dendrogram) by linking clusters based on similarity and grouping criteria. In other words, the hierarchical clustering technique initially connects the two most similar clusters and continues the process of constructing larger clusters.
|
|
9. Table 1 is needed to be rearrangement. |
Following your comment, we arranged the Table 1 based on the constructs. |

Reviewer 2 Report
In general terms, the research work entitled “Machine learning applications for the development of a questionnaire to identify Sasang constitution typology” is a research paper with high potential. However, some comments are proposed that will help to increase the scientific quality.
Introduction
The introduction in general is fine but more clear information on the following should be provided
a) How this paper will fill the gap in theory and practice
b) What unique knowledge do authors like to add to the existing literature, and what innovations bring this manuscript to scientific studies?
Theoretical background
In the theoretical background, would be useful to know more information about the general interest of the topic.
The paper should be enriched with the theoretical approach. The paper could benefit from more up-to-date references concerning the time frame 2020-22.
Methodology
The sampling frame and its advantages and limits must be clarified. When stating " Among the 225 questions, 133 were about behavioral and psychological characteristics, and 92 were related to personality characteristics " reference should be made to authors who have considered some of the questions in similar research. This should be indicated in the theoretical framework and will also facilitate a better discussion of the results.
The dendogram must be explained in depth, in this case, theory must be introduced in each of the clusters.
Results.
It is the most powerful part of the article, so different graphs and tables are used that make visual what is commented in the text. Congratulations to the authors for the development of this research.
Conclusion and policy implications
The content of the conclusions should be broader. In addition, implications and future lines of research should be included. These implications should be clear and concise so that they are really useful for organizations and researchers.
Author Response
|
Comments from reviewers |
Responses |
|
In general terms, the research work entitled “Machine learning applications for the development of a questionnaire to identify Sasang constitution typology” is a research paper with high potential. However, some comments are proposed that will help to increase the scientific quality. |
Dear Reviewer, we appreciate your time and effort in reviewing our paper. We made every effort to answer thoroughly to all of your remarks. Please see our responses to your comments listed below. Thanks. |
|
Introduction The introduction in general is fine but more clear information on the following should be provided a) How this paper will fill the gap in theory and practice
|
We address this comment by including more literature about the study background and emphaizing the value of this study in Introduction.
[Lines 76-102] In addition, previous studies reported that Sasang constitutional identification tool needs to improve the reliability [33,34]. For example, over the course of a year, Bae, Kim, Go, Park, Lee and Lee [34] administered the same Sasang constitutional diagnostic questionnaire to university students twice. Despite no significant physical changes (i.e., BMI result), nearly one-third of the participants had different Sasang constitutional type results from the previous year, indicating the need for a more reliable questionnaire. Similarly. Lee, Yim and Kim [33] conducted test-retest reliability test with SCAT over the four weeks and found some sub-constructs of the tool had low correlation coefficient. Therefore, developing a reliable, user-friendly, and integrative tool for the Sasang-type determination is necessary. Although self-reported surveys are widely used for the Sasang-type determination, this approach may result in several biases, such as social desirability [35], measurement [36], and recall [37] biases. Measurement bias is particularly problematic when using self-administered questionnaires for Sasang determination because the language used in the questionnaire can be confusing and difficult to answer accurately, making it difficult to distinguish different Sasang constitutional types. A machine-learning approach has recently been introduced to improve the accuracy of Sasang constitution type identification; however, such effort remains limited in the Sasang constitution medicine literature [38]. Moreover, previous studies classified the respondents into one of the Sasang constitutional types, although some Sasang constitutional types may share similar pathophysiological or psychological traits [39,40]. Hence, this study aimed to discover the Sasang determination questionnaire items and key constructs that produce significantly different responses for each Sasang constitution. For this purpose, this study tested a wide range of measurement items widely implemented in previous studies and fields and implemented a feature selection algorithm to discover significant constructs and questionnaire items. The similarities and differences across the Sasang constitutional types were identified using various clustering approaches. |
|
b) What unique knowledge do authors like to add to the existing literature, and what innovations bring this manuscript to scientific studies? |
We also strengthened our Discussion to address this point.
[Lines 295-318] Sasang constitutional medicine has a long history with an emphasis on disease prevention and personalized treatment, which is consistent with contemporary preventive medicine [10]. Sasang constitutional medicine not only considers physical and clinical characteristics for personalized medicine, but also personality traits and various behavioral characteristics (e.g., eating attitude and dietary behaviors) [52-55]. Consequently, the Sasang constitution typology can serve as a guide for healthy lifestyles and the quest of mutual respect and comprehension of the four distinct constitutions [54]. Although Sasang constitutional medicine has gained appeal among the general public, the validity and reliability of Sasang constitution determination tools remains a source of concern. In addition, previous studies concerned with the development of a reliable Sasang constitution typology identification tool tended to place an excessive amount of emphasis on classifying the constitutional groups based on the origin of the Sasang constitutional medicine, Donguisusebowon, which was published at the end of the nineteenth century [52]. Consequently, it is necessary to evaluate the efficacy of numerous sub-constructs of the Sasang constitution typology using cutting-edge approaches. In order to address this issue, this study compiled previously utilized sub-constructs and questionnaire items, as well as recently validated/suggested sub-constructs and questionnaire items, in order to determine the most effective sub-constructs and questionnaire items for distinguishing Sasang constitution typologies. In addition, different rephrased questionnaire items were evaluated to determine which wording or phrases are more effective for identifying Sasang constitution type to improve the usability of the Sasang typology identification in many industries. Most importantly, this study employed clustering methods to discover pathophysiological and personality traits that are specific to each Sasang type, as well as overlapping traits that are applicable to several Sasang types. |
|
Theoretical background In the theoretical background, would be useful to know more information about the general interest of the topic. |
According to your comment, we revised Introduction section, as below. Please see the revised version of our manuscript. Thank you.
[Lines 46-93] With the growing popularity and familiarity of Sasang constitutional medicine among Koreans, various industries other than medicine have paid attention to its utility. In a previous study with 839 participants, for example, 88.4% of the surveyed reported that it is crucial to choose food ingredients that are appropriate for their Sasang typology for disease prevention and treatment [11]. Moreover, Sasang constitutional medicine has sparked interest in psychology [12], media [13], fashion [14,15], exercise [16,17], and animation [18], wine recommendation system [19]demonstrating its interdisciplinary potential in the era of the personalized industry. To expand international use of Sasang typology, Sasang constitutional medicine has been applied to people from countries other than Korea, including Chinese [20], Americans [21], and Vietnamese [22]. To further increase the usability of Sasang constitutional medicine, it is necessary to develop an objective and reliable tool that can be used to readily identify Sasang constitutions. The most reliable method for determining the Sasang constitution is to monitor improvements in symptoms with herbal medicine prescribed by Korean medicine doctors [23]. However, this method may not be easily adaptable to the Sasang Constitution in various industrial domains. The Korea Institute of Oriental Medicine has developed the Sasang Constitution Analysis Tool (SCAT) to identify an individual's Sasang constitutional types through four categories: facial, voice, body type, and questionnaire [24]. The Questionnaire for Sasang Constitutional Classification II (QSCCII) is another popular form of the Sasang Constitution Questionnaire that consists of multiple constructs, such as body shape, health status, and eating behavior [25,26]. However, concerns have been raised about how accurately these methods determine Sasang constitutional types [27,28]. To implement holistic Sasang constitution determination methods, experts in the private sector have also developed techniques using pendulum and O-ring tests employing various means such as scent oils, color cards, gold and silver rings, and observation of individual faces and body shapes [29]. Previous studies [29-32] that employed both methods mentioned above (i.e., SCAT and field experts) revealed that the two methods produced different Sasang determination outcomes, demonstrating a low concordance rate between the two methods. In addition, previous studies reported that Sasang constitutional identification tool needs to improve the reliability [33,34]. For example, over the course of a year, Bae, Kim, Go, Park, Lee and Lee [34] administered the same Sasang constitutional diagnostic questionnaire to university students twice. Despite no significant physical changes (i.e., BMI result), nearly one-third of the participants had different Sasang constitutional type results from the previous year, indicating the need for a more reliable questionnaire. Similarly. Lee, Yim and Kim [33] conducted test-retest reliability test with SCAT over the four weeks and found some sub-constructs of the tool had low correlation coefficient. Therefore, developing a reliable, user-friendly, and integrative tool for the Sasang-type determination is necessary. Although self-reported surveys are widely used for the Sasang-type determination, this approach may result in several biases, such as social desirability [35], measurement [36], and recall [37] biases. Measurement bias is particularly problematic when using self-administered questionnaires for Sasang determination because the language used in the questionnaire can be confusing and difficult to answer accurately, making it difficult to distinguish different Sasang constitutional types. A machine-learning approach has recently been introduced to improve the accuracy of Sasang constitution type identification; however, such effort remains limited in the Sasang constitution medicine literature [38] |
|
The paper should be enriched with the theoretical approach. The paper could benefit from more up-to-date references concerning the time frame 2020-22. |
To addres this comment, we updated our references as below.
Sung, K.-h.; Ryu, G.-h.; Yun, D.-y. Sasang Constitution Analysis and Wine Recommendation App suggestion through Mobile Face Recognition. International Journal of Internet, Broadcasting and Communication 2021, 13, 155-162. Kim, S.-H.; Park, K.-H.; Jeong, K.; Lee, S. A Pilot Study for Applying Korea Sasang Constitutional Diagnostic Questionnaire (KS-15) to the Vietnamese. Journal of Sasang Constitutional Medicine 2021, 33, 15-23. Chae, H.; Hwang, Y.; Kim, M.S.; Baek, Y.; Jeong, K.; Lee, J.; Lee, S.; Lee, S.J. Study on the Sasang type diagnosis using objective biopsychological measures. Journal of Sasang Constitutional Medicine 2021, 33, 1-13. Kim, S.M. Physical characteristics according to Sasang constitution typology determined by Sasang Constitution Analysis Tool (SCAT) and a specialist. Journal of the Korea Convergence Society 2020, 11, 363-371. Bernardi, R.A.; Nash, J. The importance and efficacy of controlling for social desirability response bias. Ethics & Behavior 2022, 1-17. Park, S.-Y.; Park, M.; Lee, W.-Y.; Lee, C.-Y.; Kim, J.-H.; Lee, S.; Kim, C.-E. Machine learning-based prediction of Sasang constitution types using comprehensive clinical information and identification of key features for diagnosis. Integrative medicine research 2021, 10, 100668. Lee, S.; Lee, Y.; Han, S.Y.; Bae, N.; Hwang, M.; Lee, J.; Chae, H. Urinary Function of the Sasang Type and Cold-Heat Subgroup Using the Sasang Urination Inventory in Korean Hospital Patients. Evidence-Based Complementary and Alternative Medicine 2020 Kim, M.S.; Hwang, Y.; Park, S.J.; Lee, J.; Kim, J.; Lee, S.J.; Chae, H. Systematic review on the use of Sasang Personality Questionnaire in traditional Korean medicine. Journal of Oriental Neuropsychiatry 2021, 32, 167-184. Jeong, K.; Park, K.; Lee, S.; Hwang, J.-Y.; Baek, Y. Evaluation of Dietary Behaviors According to Sasang Constitution Using a Nutrition Quotient: A Korean Medicine Daejeon Citizen Cohort Study. Journal of Sasang Constitutional Medicine 2020, 32, 86-95. Kim, J.-W.; Jeon, S.-H. A Study on the Relationship between the Eight Principle Pattern Identification of Cold-Heat, Deficiency-Excess and the Sasang Constitution-500 Women with Menstrual Pain and Women without Menstrual Pain as a Target. Journal of Sasang Constitutional Medicine 2020, 32, 18-32. |
|
Methodology The sampling frame and its advantages and limits must be clarified. When stating " Among the 225 questions, 133 were about behavioral and psychological characteristics, and 92 were related to personality characteristics " reference should be made to authors who have considered some of the questions in similar research. This should be indicated in the theoretical framework and will also facilitate a better discussion of the results. |
We added references and examplary items for each consturct.
[Lines 125-137] 2.3. Measurement items For the questionnaire, 225 questions were included based on the SCAT 2.0 questionnaire by the Korea Institute of Oriental Medicine, ideological medicine books [44], and field experiences. Among the 225 questions, 133 were about pathophysiological characteristics [45-50], and 92 were related to personality characteristics [47,48]. The questionnaire items include physiological systems about cold and heat [45] (e.g., My hands are cold), body shape appearance [48] (e.g., I have a petite and slender physique), digestive functions and eating habits (e.g., I usually eat slowly) [50]. The exemplary items regarding personality characteristics are “I tend to be inactive”, “I am sociable”, “I enjoy speaking in public”. All questions were answered with 'yes (coded as 1) or no (coded as 0). This study included several similar questionnaire items on the same attribute to identify the types of questions that reduce confusion among the respondents and confirm the internal consistency of the survey responses on the same attribute.
Moreover, in Discussion, we added more theoretical implication related to the aforementioned issue.
[Lines 310-318] In order to address this issue, this study compiled previously utilized sub-constructs and questionnaire items, as well as recently validated/suggested sub-constructs and questionnaire items, in order to determine the most effective sub-constructs and questionnaire items for distinguishing Sasang constitution typologies. In addition, different rephrased questionnaire items were evaluated to determine which wording or phrases are more effective for identifying Sasang constitution type to improve the usability of the Sasang typology identification in many industries. Most importantly, this study employed clustering methods to discover pathophysiological and personality traits that are specific to each Sasang type, as well as overlapping traits that are applicable to several Sasang types.
[Lines 331-336] As a result, this study discovered pathophysiological symptoms and personality are both important constructs to identify the Sasang constitution types. For instance, questions regarding sweat and temperature were found to be effective to distinguish different Sasang types in addition to questions regarding eating habits and body shapes. Furthermore, many questionnaire items regarding personality were found to be effective for Sasang constitution identification. |
|
The dendogram must be explained in depth, in this case, theory must be introduced in each of the clusters. |
The explanation about dendrogram and graph theory were introduced in Methodology.
[Lines 163-169] Based on graph theory, the relationships between the items were represented by the hierarchical cluster dendrogram [51,52]. According to graph theory, graphs are architectures consisting of a collection of nodes and edges and that describe data relationships. Hierarchical clustering technique aims to produce a network structure (dendrogram) by linking clusters based on similarity and grouping criteria. In other words, the hierarchical clustering technique initially connects the two most similar clusters and continues the process of constructing larger clusters. |
|
Results. It is the most powerful part of the article, so different graphs and tables are used that make visual what is commented in the text. Congratulations to the authors for the development of this research. Conclusion and policy implications The content of the conclusions should be broader. In addition, implications and future lines of research should be included. These implications should be clear and concise so that they are really useful for organizations and researchers. |
Thank you for your comment. We made substantial changes in Discussion.
[Lines 294-352] 4. Discussion Sasang constitutional medicine has a long history with an emphasis on disease prevention and personalized treatment, which is consistent with contemporary preventive medicine [10]. Sasang constitutional medicine not only considers physical and clinical characteristics for personalized medicine, but also personality traits and various behavioral characteristics (e.g., eating attitude and dietary behaviors) [54-57]. Consequently, the Sasang constitution typology can serve as a guide for healthy lifestyles and the quest of mutual respect and comprehension of the four distinct constitutions [56]. Although Sasang constitutional medicine has gained appeal among the general public, the validity and reliability of Sasang constitution determination tools remains a source of concern. In addition, previous studies concerned with the development of a reliable Sasang constitution typology identification tool tended to place an excessive amount of emphasis on classifying the constitutional groups based on the origin of the Sasang constitutional medicine, Donguisusebowon, which was published at the end of the nineteenth century [54]. Consequently, it is necessary to evaluate the efficacy of numerous sub-constructs of the Sasang constitution typology using cutting-edge approaches. In order to address this issue, this study compiled previously utilized sub-constructs and questionnaire items, as well as recently validated/suggested sub-constructs and questionnaire items, in order to determine the most effective sub-constructs and questionnaire items for distinguishing Sasang constitution typologies. In addition, different rephrased questionnaire items were evaluated to determine which wording or phrases are more effective for identifying Sasang constitution type to improve the usability of the Sasang typology identification in many industries. Most importantly, this study employed clustering methods to discover pathophysiological and personality traits that are specific to each Sasang type, as well as overlapping traits that are applicable to several Sasang types. In summary, this study tested and analyzed existing Sasang constitution questionnaire items to provide future resources for developing a robust Sasang analysis tool. This study suggests meaningful industrial and academic implications by adopting various machine learning approaches that have rarely been implemented in Sasang constitutional medicine studies. First, this study conducted feature selection to determine which attributes produced different responses across Sasang constitutional types. Previous studies attempted to select key measurement items from a large set of measurement items used in prior studies and in the field to reduce survey respondents’ fatigue by answering a large number of questions and enhancing the outcomes of Sasang constitution identification [27,58]. This study proposes a new strategy to discover relevant attributes for Sasang constitution identification and improve future machine learning outcomes by implementing a machine-learning algorithm called feature selection. As a result, this study discovered pathophysiological symptoms and personality are both important constructs to identify the Sasang constitution types. For instance, questions regarding sweat and temperature were found to be effective to distinguish different Sasang types in addition to questions regarding eating habits and body shapes. Furthermore, many questionnaire items regarding personality were found to be effective for Sasang constitution identification. Second, the results of the hierarchical clustering analysis revealed that some attributes are salient to one Sasang constitution, whereas others may be applicable to several types. The findings imply that certain attributes are unique characteristics of a particular Sasang constitution, and some attributes can be found in various Sasang constitutional types. Previous research suggests that Tae-Yang and So-Yang typologies, as well as So-Yang and So-Eum typologies, share similar biological traits and, thus, exhibit comparable behavioral and psychological traits [26,59]. This study not only verifies the findings of prior research, but also illustrates the overlapping constructs and prominent characteristics of several typologies. Our findings, therefore, can be useful to include attributes that represent the distinct characteristics of each Sasang constitution to develop a timely and effective questionnaire, especially when the Sasang constitution questionnaire is used as a supplementary tool along with other methods. However, including measurement items applicable to several types can be useful in developing a comprehensive Sasang constitution questionnaire. In this case, the results should be interpreted based on understanding the relationship between each question and Sasang constitution, rather than simple summary methods, such as the sum or average of all item results).
|

Round 2
Reviewer 1 Report
In this version, the authors have added some experimental results and modifications to respond positively to my questions.
Author Response
We greatly appreciate your time and effort to review our paper.